# Opening decision spaces: A case study on the opportunities and constraints in the public health sector of Mpumalanga Province, South Africa

**Sophie Witter**[1], **Maria van der Merwe**[2,3], **Rhian Twine**[3], **Denny Mabetha**[3], **Jennifer Hove**[3], **Stephen M. Tollman**[3], **Lucia D'Ambruoso**[4]*

**1** Institute for Global Health and Development, Queen Margaret University, Edinburgh, Scotland, United Kingdom, **2** Independent Consultant, White River, South Africa, **3** MRC/Wits Rural Public Health and Health Transitions Research Unit (Agincourt), School of Public Health, University of the Witwatersrand, Johannesburg, South Africa, **4** Aberdeen Centre for Health Data Science, Institute of Applied Health Sciences, School of Medicine, Medical Sciences and Nutrition, University of Aberdeen, Scotland, United Kingdom

\* lucia.dambruoso@abdn.ac.uk

**Data Availability Statement:** The main datasets used and/or analysed during the current study are in the public domain in the form of (a) referenced

## Abstract

### Background

Decentralised and evidence-informed health systems rely on managers and practitioners at all levels having sufficient 'decision space' to make timely locally informed and relevant decisions. Our objectives were to understand decision spaces in terms of constraints and enablers and outline opportunities through which to expand them in an understudied rural context in South Africa.

### Methods

This study examined decision spaces within Mpumalanga Province, using data and insights generated through a participatory action research process with local communities and health system stakeholders since 2015, which was combined with published documents and research team participant observation to produce findings on three core domains at three levels of the health system.

### Results

Although capacity for decision making exists in the system, accessing it is frequently made difficult due to a number of intervening factors. While lines of authority are generally well-defined, personal networks take on an important dimension in how stakeholders can act. This is expressed through a range of informal coping strategies built on local relationships. There are constraints in terms of limited formal external accountability to communities, and internal accountability which is weak in places for individuals and focused more on meeting performance targets set at higher levels and less on enabling effective local leadership. More generally, political and personal factors are clearly identified at higher levels of the

reports and papers, upon which secondary analysis was performed, and (b) published VAPAR papers, which report on associated datasets. All project outputs which are referenced and supporting resources can be found at https://www.vapar.org/, including presentations which were used for this analysis. Other sources, such as observational notes, are not suitable for sharing without contextualisation.

**Funding:** The research presented in this article is funded by the Health Systems Research Initiative from Department for International Development (DFID)/Medical Research Council (MRC)/Wellcome Trust/Economic and Social Research Council (ESRC), https://www.ukri.org/what-we-offer/browse-our-areas-of-investment-and-support/health-systems-research-initiative/, grant number MR/P014844/1, recipients (SW, LD, MV, ST, RT). All views expressed here are those of the authors alone. The sponsors or funders did not play any role in the study design, data collection and analysis, decision to publish, or preparation of the manuscript.

**Competing interests:** The authors have declared that no competing interests exist.

**Abbreviations:** APP, Annual Performance Plan; CEO, Chief Executive Officer; DCST, District Clinical Specialist Teams; DoH, Department of Health; HDSS, Health and socio-demographic surveillance system; MCWY&H, Maternal, child, women and youth health and nutrition; PAR, participatory action research; PFMA, Public Finance Management Act; PHC, primary health care; TO, team observation; VA, verbal autopsy; VAPAR, Verbal Autopsy with Participatory Action Research.

system, whereas at sub-district and facility levels, the dominant theme was constrained capacity.

## Conclusions

By examining the balance of authority, accountability and capacity across multiple levels of the provincial health system, we are able to identify emergent decision space and areas for enlargement. Creating spaces to support more constructive relationships and dialogue across system levels emerges as important, as well as reinforcing horizontal networks to problem solve, and developing the capacity of link-agents such as community health workers to increase community accountability.

## Introduction

Decentralisation is a widely adopted and promoted strategy as a means towards health sector reform [1]. In South Africa, the health system has been administratively decentralised since 1994 to provincial level, consistent with the overall policy for government in the South African Constitution [2]. Under the National Health Act of 2003, provincial departments of health are mandated to provide healthcare services. The provincial health authority is responsible for the management of the provincial heath budget and delivering all health services and to adapt national policies according to the needs of the province. The majority of the South African population access health services through the public sector district health system, which is the preferred government mechanism for health provision within a primary health care approach. District and sub-district health management offices oversee management of the primary health care (PHC) facilities (clinics, community health centres and district hospitals), in line with core national standards and towards achieving targets set largely at the provincial level for key population health indicators.

On the health system side, this approach relies on there being decision space, meaning decision-making power which can be exercised by managers [3] within public administrations to act on better evidence of community challenges and priorities. The aim of this article is to examine this in practice at multiple levels in one province in South Africa, using the Verbal Autopsy with Participatory Action Research (VAPAR) project data and insights (www.vapar.org), participant-observation by team members, public documents and public data. Its objective is to understand constraints on decision space but also opportunities to expand and enhance it, with a particular focus on the role of evidence and co-production. The article focuses on child health as a tracer condition with which we have engaged, although many of the features highlighted are cross-cutting to other programme areas. It adds to existing literature by using a decision space lens at multiple levels of a decentralised system and focusing on a broad range of capacities, some of which (such as infrastructure and information) have been relatively neglected in the literature to date [4].

## Materials and methods

### Definitions and framework

Decision space is defined as "the range of effective choice that is allowed by the central authorities to be utilised by local authorities" [3] and represents the degree of decentralisation granted to an individual or organisation. This space can be formally defined by laws and regulations,

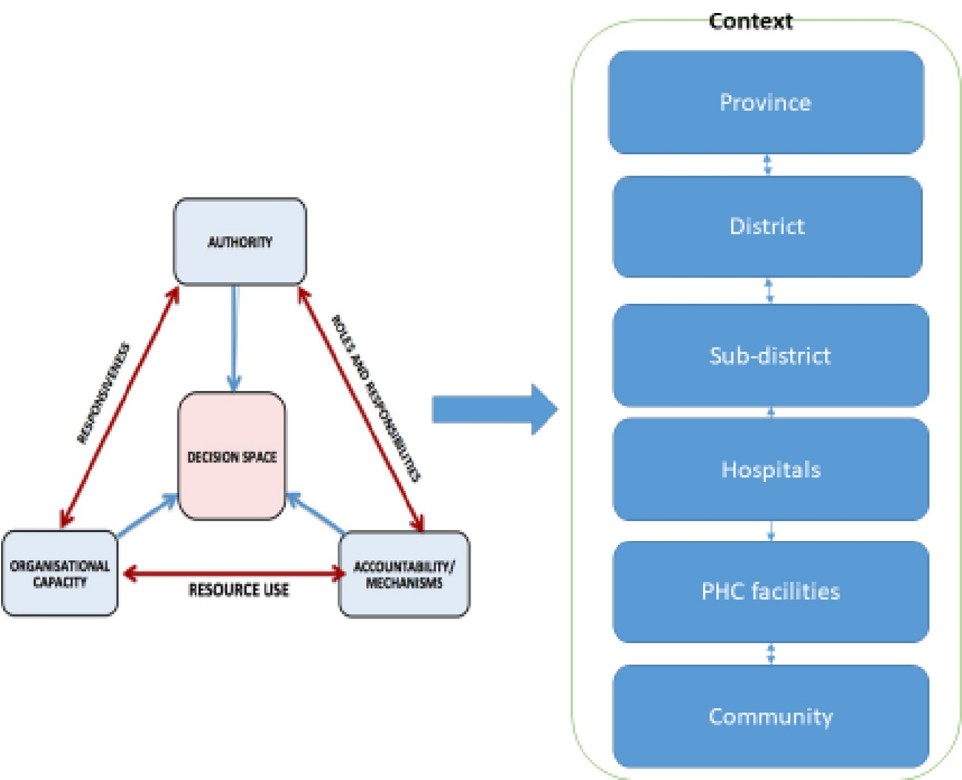

**Fig 1. Decision space model.**

or informally by lack of enforcement of these formal definitions that allows lower level officials at each level to 'bend the rules'. The assumption is that with increased decision space, managers can make decisions that are more innovative, efficient and responsive to local conditions and that this will improve the quality of service delivery [4].

The paper adopts a conceptual framework (Fig 1) which sees decision space as an emergent property of the authority, capacity and accountability within an organisation, as well as the context in which it operates. Authority provides the *de jure* decision space by defining roles and responsibilities which enable managers and staff to take action. However, while it is necessary, it is not sufficient, if other features (capacities and accountability) are missing or misaligned. When authority is unclear or fragmented, as is commonly the case, accountability can be undermined [5].

*Source*: *adapted from* [5]

Capacity is what enables the organisation to function [6], and includes administrative, technical, organisational, financial and human resources. Some of these capacities are inputs, such as finance and staff and valid health system data, but they also include the capacity to manage inputs well. Resources (inputs) are necessary but not sufficient without management capacity, with which they are interdependent [5]. Without resource capacities to ensure organizational functioning, management capacities could also be redundant. Capacity and authority also need to go in step if resources are to be used well. Limited evidence suggests that where resource capacities fall short of what is required for organizational functioning, managers may resort to informal decision-making strategies to fulfil responsibilities and mitigate bureaucratic constraints [4].

Desirable features of capacities link closely to standards for strong health systems generally. For example, financial capacity would be assessed according to adequacy, regularity, flexibility and predictability [7]. For human resources, control over hiring, payment, performance assessment and management, and motivation would be key. For management, technical skills such as planning and management of staff, medicines and supplies and infrastructure are important, as well as having access to relevant information and the ability to use it effectively. Health management information systems produce large amounts of data, yet data are rarely used in local decision-making [8].

In addition, leadership capacity is critical–the ability to create and share organisational vision and motivate staff to adopt it, for example [9]. According to Gilson et al. (2014), PHC will only become a lived reality within the South African health system when front line staff are able to make sense of policy intentions and incorporate them into everyday routines and practices [10]. This requires a leadership of sensemaking that enables front line staff to exercise collective discretionary power to deal with particular contextual needs at the front line of policy delivery.

Accountability refers to answering for decisions or actions, often with the purpose of reducing abuse and improving system performance. It can refer both to the mechanisms for being held responsible for doing the right things and doing them effectively [5, 6], but also the outcome of these. Commonly, a distinction is drawn between bureaucratic and external accountability [11]–the former indicating vertical systems within an organisation, such as planning, target setting, supervision, monitoring, reporting and audits, while the latter is most commonly enacted in the health system through formal community participation mechanisms such as health facility committees and hospital boards. Accountability affects decision space (positively and negatively) and helps to direct its use [12].

Community accountability [13] is an approach to strengthening public accountability through direct involvement of clients, users or the general public in health service delivery. Different measures to enhance community accountability, linked to peripheral facilities, include clinic committees and health interest groups, public report cards and patients' rights charters. A recent literature review highlighted the way in which bureaucratic accountability mechanisms often constrain the functioning of external accountability mechanisms. Front line managers and providers may be constrained from responding to patient and population priorities due to organizational cultures characterised by supervision and management systems focused on compliance to centrally defined outputs and targets [14]. This may be driven by the need to control misuse of resources, lack of confidence in managers' capacities, or both. This bureaucratic accountability can crowd out community accountability [15], creating a 'compliance culture' which focuses more on tasks than outcomes.

The role of context is also widely recognised as important in relation to these domains and their impact on decision space. Socio-cultural and political factors influence all these relationships, which together affect roles and responsibilities in the health system, its responsiveness and how resources are used [5].

## Setting

In post-apartheid South Africa, there was a constitutional commitment to the right to health and community participation for PHC to overcome the historical disadvantages faced by the majority population [16]. Today, significant pro-poor, equity-oriented reforms include: National Health Insurance [16, 17]; PHC Re-engineering [18], which includes the development of Ward-Based Primary Healthcare Outreach Teams decentralising PHC to community level; and the Ideal Clinic initiative, which provides a national quality framework within PHC

Re-engineering. Successive national annual performance plans also address workforce development and planning with initiatives on affirmative student recruitment (prioritising students from disadvantaged backgrounds), financial incentives, foreign recruitment and compulsory post training service as well as commitments to strengthen the public sector health workforce through National Health Insurance and the National Development Plan [19].

While policy development is progressive and inclusive, significant gaps exist between policy and implementation in a system characterized by chronic underinvestment, human resource crises, widespread corruption, poor stewardship and deteriorating infrastructure [20, 21]. Health systems also faces a complex 'quadruple' burden of socially patterned mortality comprising chronic infectious diseases (HIV/AIDS and TB), non–communicable conditions, maternal and child mortality, and mortality owing to injury and violence [22–25]. The burden of HIV is high and highly unequal. Prevalence in black populations is 40–50 times that of white populations and in adolescents, risks are eight times higher in females than males [20]. South Africa's child poverty rate is relatively high and its Gini coefficient at 63 is the highest globally, with the majority black population remaining disadvantaged [25]. While early on in the pandemic, South Africa was internationally recognised for timely and decisive action in response to Covid-19, it remains one of the most unequal countries globally, and this is likely to be both reflected in and exacerbated by the pandemic.

Mpumalanga is one of nine provinces in the country, with a population of 4.7 million (7.9% of the national population [26] and a gross domestic product per capita of $12,585 that is close to the national average but well below provinces such as Western Cape and Gauteng [27]. More than half the population is rural [28], while nationally this is around one third [25]. In 2019, provincial unemployment was 35%, with 51% living in poverty [29]. In 2015, life expectancy for males and females was 50 and 53 years respectively, lower than the national average of 60 and 67 years, and under-5 mortality was 41 deaths per 1,000 live births in 2012, which is comparable nationally [30–32].

Mpumalanga Department of Health (DoH) has responsibility to deliver need-based services through an integrated health system covering three districts and 17 sub-districts [30, 33], which include 279 clinics and 33 hospitals [34]. The Mpumalanga DoH structure includes five strategic directorates: HIV/AIDS, sexually transmitted infections and TB Control; communicable disease control; non-communicable diseases; maternal, child, women and youth health and nutrition (MCWYH&N); and research and epidemiology.

Verbal autopsy (VA) data from MRC/Wits-Agincourt Unit's health and socio-demographic surveillance system (HDSS) in Mpumalanga shows child mortality to have recently declined to similar levels as in the early 1990s. Although this may not seem like a major achievement, on closer examination it clearly has been so, in terms of reversing the disastrous effects of the HIV epidemic on mortality patterns during a period of emerging democracy following decades of apartheid [35, 36]. There are a number of national policies targeting child health, such as the Child Healthcare Problem Identification Programme, which uses child death audits to identify and address immediate and root causes, and the Integrated School Health Policy advancing integrated, holistic approaches to school health [36]. Further progress towards Sustainable Development Goal targets will require widespread improvements in socio-economic conditions and health systems functionality, with national and local leadership commitment.

Within Mpumalanga, we have focused on Ehlanzeni district and Bushbuckridge sub-district, as these are the sites within which the HDSS is located. Ehlanzeni district, the largest of three districts in Mpumalanga with an estimated population of 1.8 million [37]. The district is situated in the south eastern part of the province, bordering Mozambique and Swaziland and consists of four local (sub-district) municipalities.

## Design

This paper adopts a qualitative synthesis design [38],drawing on analysis of mixed data sources from 2015 to 2020, which are combined to investigate the domains of our conceptual framework. The first strand draws from the VAPAR programme, which is a partnership of the MRC/Wits-Agincourt Research Unit, Mpumalanga DoH, and collaborating researchers dating back to 2015, to connect community-generated evidence to practitioners, planners and managers through a coproduction, action research process. The second strand draws from experiences of VAPAR team members, and the third from reports, data and documents on health service delivery in the province.

## Data sources

The VAPAR process combines VA, a method to determine levels and causes of death in settings where deaths go unrecorded, and participatory action research (PAR), a process in which different stakeholders organise evidence for action. A pilot followed by a series of cycles of PAR was progressed, comprising observation, analysis, planning and action stages.

Community stakeholders were engaged in the initial observation stage from three rural villages, selected based on demographic variation and feasibility, with total populations of between 4,000 and 6,500, within the MRC/Wits-Agincourt Unit's HDSS study area [39]. Relationships with community stakeholder groups were developed through a series of community-based workshops in which we coproduced new knowledge on under-5 deaths, alcohol and drugs, and water using methods such as ranking, diagramming, stakeholder mapping and participatory photography [40]. A total of 48 community stakeholders participated from three purposively selected villages [41]. In addition, data from the Agincourt HDSS, including VA data on causes and circumstances of death, were integrated into the process [42].

To analyse VA and PAR data and plan feasible action, we held a series of further workshops engaging broadly with government departments, non-governmental agencies and community stakeholders. We jointly identified actions, timescales and implementing partners, culminating in a joint Local Action Plan that all participants committed to. We collected data in presentations, registers, minutes, observational notes and reflective journal data to develop accounts of the process, as well as substantive interpretations and proposed responses.

In the final phase of the cycle, actions were monitored, with cooperative reflection and learning feeding into the following cycle. Those who had committed to specific action items were visited by the researchers at venues of their choice to discuss progress. A structured tool captured mechanisms of change. The process culminated with a collective reflection. Using rapid, participatory methods, we conducted interviews with participants from local communities, government departments and parastatals, non-governmental organisations, and held two workshops with health systems actors. One workshop was held at provincial level with programme and directorate managers, and one at national level, with health programme and policy specialists. We sought perspectives on whether and how impacts had been achieved; acceptability and utility of the process; levels and mechanisms for integration into the health system; and future linkages [41].

In addition to VAPAR data, this paper draws from team discussions which reflected on and elicited the experiences of team members, many of whom have extensive experience of working with and in embedded research environments and different levels and sections of the health system over a number of years, and whose engagement in the VAPAR process has allowed for insights into the receptivity of the system to research evidence as well as insider/outsider perspectives on systems functioning across all the health system blocks.

We also draw on public documents such as provincial and district plans, expenditure reports, media reports, other relevant evidence generated by the MRC/Wits-Agincourt Unit and relevant recent reports on the health system in Mpumalanga produced by other organisations; these were sought using key words relating to the system blocks and levels, combined with locality terms, and selected purposively to inform elements of our framework, alongside wider global and health system research from South Africa. Data or insights were drawn from 78 documents, of which eight related to the VAPAR project, 12 provided evidence on Mpumalanga province, 42 related to the wider health system in South Africa and 16 discussed decision space in other settings and conceptually.

## Data analysis

Analysis was undertaken using the decision space framework and integrating data from the main strands by the research team (Table 1), initially in a research team workshop in 2019, and subsequently through iterations of the paper and subsequent reflective research team meetings. The research team conducted a rapid literature review on decision space in health care and identified a framework [5] to structure its analysis. Literature was shared with team members and discussion held on its application to the local context. Data were factual (commonly from official sources, such as provincial reports) or interpretative (thus representing important views of sectoral performance), so quality assessment of these sources was not applied or relevant. Analysis was abductive, focused on patterns within domains of the conceptual framework, comparing across levels and reaching consensus by team discussions and drafting. As decision space is not observed directly, but is emergent from the domains assessed, team interpretation was applied to derive a commentary on decision space.

## Ethical considerations

This article draws from the PAR data. For the PAR, informed consent was sought from all participants, all identifiable data were anonymised, and approvals were obtained from the authors' institutes and from the provincial health authority. Additional information regarding the ethical, cultural, and scientific considerations specific to inclusivity in global research is included in the Supporting Information (S1 Checklist).

## Results

In the findings, we examine the extent and inter-relationship of authority, accountability and capacity in the local levels of the health system (facility, district and subdistrict). This is followed by a provincial account of constraints and enablers to decision space. Tables 2 and 3 provide an overview of findings.

**Table 1. Summary of data sources.**

| Type of data | Selection and analysis |
| --- | --- |
| Existing VAPAR programme data | Re-analysis of programme data (workshop reports and outputs; VAPAR papers; earlier participant interviews), using research framework |
| Team observations (TO) | Experiential knowledge of team, recorded during team workshops, structured by conceptual framework domains (marked TO in findings below) |
| Official reports | Recent national, provincial and district reports and media reports, purposively selected and with data extracted to populate framework domains |
| Secondary research literature | Searched using key search terms linked to decision space; this literature helped shape the paper's approach and the contextualisation and discussion of the findings. Publications on recent studies on PHC services in the area were also used. |

**Table 2. Summary of findings, by domain.**

| Level | Authority | Accountability | Capacity | Emergent decision space and its implications |
|---|---|---|---|---|
| Facilities | Clear roles within hospitals and community health centres, including specialist child health posts at district hospital level | Formal accountability structures exist but often lack capacity and de facto power. Parallel channels of accountability emerging, including litigation and community protests. Linking staff, such as CHWs, exist to connect facilities and communities, but they are under-resourced in relation to their tasks. Vertical accountability for programmes is fragmented. | Multiple challenges, including insufficient staff, inadequate competencies, poor staff motivation and attendance, poor working conditions, lack of maintenance and functionality of facilities, gaps in transport and supplies, incomplete information systems. | Decision space exists, though constrained by resources and culture of bureaucratic accountability. High expectations without sufficient resourcing and mixed signals on accountability can reduce responsiveness but we also find evidence of positive informal coping strategies (e.g. sharing of supplies). |
| District/ subdistrict | Clear roles for district health management teams, programme directors and PHC coordinators at sub-district level, albeit there is some potential duplication between supervisory and technical support roles. | Strong upward formal accountability, but can be focused on fault-finding. Vertical communication depends on personal factors; limited space in system for dialogue across levels around problem-solving. | Centralised (to provincial level) hiring of staff and procurement of supplies and routine services (e.g. maintenance) limiting operational effectiveness at district level. Infrastructure is limited and poorly distributed, skewing use towards secondary services. Concerns over staff safety and poor working conditions. | The district and sub-district have an important role in operationalising policy, but vertical accountability can be punitive, rather than supportive, and resource constraints are commonly cited. However, good personal relations and communication can ease day to day functioning. During COVID, local intersectoral action was able to be taken in response to the crisis. |
| Provincial | Formal roles are clear but are fluid in practice, depending on personal networks; prevalence of 'acting' positions can create authority vacuum | Clear upward accountability in theory, but not enforced for many programme areas. Downward accountability limited by top-down planning and budgeting and unreliable resourcing of work plans. | Low expenditure per capita, poor distribution and use of funds, and mismatch of resources to policy initiatives is noted. Hiring gaps and some non-merit-based appointments. Research and data culture is limited but growing. Poor management of procurement contributing to supply gaps and inefficiencies/poor quality. | Decision space determined by personal and relational factors, more than formal authority, accountability or capacity. This poses a risk at this level to performance of the health system. |
| **Summary** | Ambiguity over roles decreases at lower system levels; this may be typical as there is less complexity and less role for patronage at operational levels | Clear formal accountability at all levels, however, informal channels are increasingly used, which may express a lack of trust in formal system responsiveness. | Capacity gaps and rigidities in resource deployment are most obvious at operational levels, which reflect constraints across all levels. | Creating spaces to support more constructive relationships and dialogue across system levels emerges as important, as well as reinforcing horizontal, peer-to-peer networks to problem solve at local level, and developing the capacity of link-agents such as CHWs to increase community accountability. |

### Facility level

At facility level, we find that decision space exists, though it is constrained by lack of resources and a culture of bureaucratic, often fragmented, upward accountability. Decision space is demonstrated by facilities which use positive informal coping strategies (such as sharing of supplies across facilities to overcome shortages) and evidenced by the range of performance by clinics in similar settings in terms of waiting times, for example.

### Authority

Key roles are clearly defined in facilities. These include, in hospitals, the Chief Executive Officer (CEO) and a Management team, including a Clinical Manager and Nursing Manager, who are in-charge in hospitals, alongside (mainly in tertiary and regional hospitals) specialist staff dealing with child health, such as hospital-based paediatricians. Within clinics, Operational Managers (usually professional nurses) are in-charge.

**Table 3. Summary of findings: Roles, constraints on these, and implications for system performance.**

| LEVEL & DOMAIN | EXPECTED ROLE (based on public policy documents) | FACTORS LIMITING DECISION SPACE (from study findings) | IMPLICATIONS FOR SYSTEM PERFORMANCE (from study findings) |
|---|---|---|---|
| **Facilities** | | | |
| Planning & budgeting | Represented during district planning process and target setting | - Vertical directed prescripts lack local relevance<br>- Program policies in prescripts not integrated and competing for prioritisation<br>- Budget and resource allocation, as well as training and development of employees, directed from higher level and not in line with operational needs | - Objectives and targets not realistic and achievable<br>- Local community needs not addressed<br>- Dissatisfaction with work conditions<br>- Overburdened operational staff<br>- Overcrowding and long waiting times |
| Implementation | - Provide comprehensive primary (clinics and community health centres) and secondary (district hospitals) health care services to communities | - Disconnection between PHC facilities and district level hospitals<br>- Disconnection between DCST, program and facility priorities | - Disconnect in service delivery, including poor implementation of referral pathways<br>- Disruption of access to appropriate level of care<br>- Skewed utilisation of health care services at different levels of care |
| Reporting & accounting | - Internal quarterly review and represented at district annual review | - Health information system provides limited information needed for facility management<br>- Corrective actions for poor reporting and performance identified at district / sub-district or provincial level, without consultation with operational teams or linking to resource availability | - Social and other underlying health determinants not identified and addressed<br>- Operational information of interest to facility managers not well reported and used |
| **District / sub-district** | | | |
| Planning and budgeting | - District health planning in preparation for and following provincial strategic planning<br>- Guide operations for priority programmes and identify programmatic needs<br>- Advise on clinical governance and quality and measures taken<br>- Review budgets and respond to procurement needs<br>- Facilitate recruitment prioritisation for vacant posts to ensure maximum impact for available resources | -High vacancy rate and centralised administrative processes.<br>- Priorities and targets set at national and / or provincial levels, rather than being based on operational priorities and capacity | - Restricted ability to plan<br>- Objectives and targets not realistic and achievable<br>- Deficits in competencies at operational level<br>- Lack of support to and motivation of operational stakeholders |
| Implementation | - Compile and oversee implementation of district operational plans | - Wide span of responsibility with limited resources<br>- Downward links from program managers to facilities often depending on personal relationships | - Inequality in performance between health facilities<br>- Lack of technical support from program leads |
| Reporting and accounting | -Internal quarterly and annual review of performance, followed by review with provincial management<br>-Raise challenges experienced by clinics, propose interventions and obtain commitment from management to address bottlenecks<br>- Reporting of performance and status of implementation of actions | - Program managers report to district PHC managers, with no accountability to provincial program managers, potentially leading to misalignment of priorities between different districts or sub-districts<br>- Performance data presented collectively and not per individual facility. | - Strategies and targets lack local relevance<br>- Clinical operations disrupted by infrequent and demotivating support to facilities<br>- Performance variation between facilities not noted and addressed |
| **Province** | | | |
| Planning and budgeting | - Provincial strategic planning, including target setting for performance indicators<br>- Management of provincial health budget<br>- Adapt national policy to the needs of the province | -Limited accountability due to top-down approaches | - Objectives and targets not realistic and achievable<br>- Poor distribution of resources |

*(Continued)*

**Table 3.** (Continued)

| LEVEL & DOMAIN | EXPECTED ROLE (based on public policy documents) | FACTORS LIMITING DECISION SPACE (from study findings) | IMPLICATIONS FOR SYSTEM PERFORMANCE (from study findings) |
|---|---|---|---|
| Implementation | -Individual programmes / units support activities to achieve targets | -Routine reporting mechanisms often focused on problems, based on sometimes questionable data reported through the information system, rather than identifying, understanding, supporting and enabling leadership, supervision and innovation | - Lack of technical support by program leads<br>- Insecurity and instability |
| Reporting | -Provincial quarterly and annual review of performance | - The district health information system only provides quantitative data and there is mostly no account of social factors and contributors. | - Strategies and targets lack local relevance<br>- Inadequate service organisation and infrastructure |

Sources (for expected roles): (17), (59)

## Accountability

At hospital level, the CEO and hospital management team report to hospital boards, which oversee the work of the facility. Hospital boards have power (including to go to provincial leadership with problems) but tend not to use these mechanisms and are often focused on political priorities (TO). Although hospital boards do receive financial reports, very few take responsibility for hospital financial matters. As a result, Boards can lack capacity (see below). The Portfolio Committee report 2017/18 reports constraints to functionality in some hospital boards and that community protests have resulted in the removal of some CEOs, indicating a form of informal accountability claimed by communities [34].

Litigation is becoming more common as a strategy by families (and lawyers) to hold the DoH to account for failures, though this mainly has consequences for provincial budgets more than for individual facilities or practitioners. In 2017/18, there were 61 legal cases outstanding, mostly relating to maternal deaths [34]. At a clinical level, audits and adverse event reporting and committees are part of the quality assurance process, although these committees vary in degrees of activity and effectiveness (TO).

At primary level, community health centre or clinic committees are the main formal accountability structure. Committees should be made up of elected community representatives and health professionals to allow community concerns to be heard and addressed. Various mechanisms are prescribed to support engagement with and feedback from communities, including complaints systems, satisfaction surveys and waiting times reporting, but these are not seen as comprehensive. In addition, clinics are required to hold Open Days as part of the Ideal Clinic initiative, however from our participant observation these have limited attendance and substance, mainly as communities are generally already aware of services rendered by clinics. While home-based carers, community health workers, and Ward-based PHC Outreach Teams are visible links between communities and facilities, they face many demands, limited recognition in the system, and no additional resources to support a rapidly expanding mandate. In this context, clients tend to use direct action such as protests or the media (including increasingly social media) rather than formal channels to address grievances. These are problematic, imposing system costs and do not represent balanced or constructive input from the community.

In terms of vertical accountability, facilities have to organise services within the policies passed down to them for each programme, such as child health. VAPAR participants perceive policies, programmes, audits and other initiatives prescribed from higher levels as well-intentioned but insufficiently tailored to local conditions and resources. Furthermore, many parallel

initiatives were described as having an overall destabilising effect on over-burdened health workers, compounding problems in an already constrained system. As a result, objectives are often not achieved and results are manipulated to show progress. This was described as '*changing numbers rather than really affecting change*' [42]. Clinics are cost centres and can plan and manage day to day operations, however budgets are set at higher levels and major inputs (drugs, supplies, staffing, equipment) are procured and supplied by higher system levels. There is no longer a direct supervisory link between district hospitals and the Community Health Centres or clinics, which are supervised by PHC supervisors (professional nurses), based in the sub-district.

## Capacities

Accountability is closely linked to capacities, which include resources of various kinds as well as their management. Reflecting on challenges in relation to under-five mortality, VAPAR participants highlighted insufficient and absent health workers, deficits in competencies (e.g. gaps in Integrated Management of Childhood Illness training, or in management of severely malnourished children and health education), support and motivation as contributory factors in under-5 deaths, and which collectively limit quality and foster poor attitudes more generally [42]. Responding to shortages, some staff act outside their mandated professional scope. Examples of strong personal commitment are seen in some cases, this is contrasted by poor management of patients in other cases [43–45]

A recent assessment in nine clinics in the study area identified issues with staff vacancies, with 55% reporting one or more vacancy, and some services affected by staff action over remuneration [46]. Nevertheless, in the same study, providers were highly satisfied with the nature of their work and generally with relationships with colleagues, but reported dissatisfaction with working conditions in terms of staffing, supplies and space. Staff also raised concerns about frequent absences of colleagues, lack of support staff, poor motivation, as well as infrequent and demotivating supervision and trainings which distract from clinical care. Additionally, some providers were less satisfied with opportunities for and quality of training and supervision and an average of 42% of providers (8% to 81% by clinic) reported plans to leave in the next two years.

Inadequate service organisation and infrastructure were further challenges identified by VAPAR participants. Specific issues included overcrowding, especially during morning clinics, exacerbating staffing problems and causing long waiting times and delayed or postponed consultations. Lack of consulting rooms and water outages were also cited: many clinic buildings date back to when the area was a homeland (areas designated under the Apartheid regime for African self-government) and significantly smaller populations were provided with limited and poor quality services. Public facilities now cater for much larger populations and are competing for infrastructure budget with other projects, resulting in higher throughput than is manageable [39].

The clinic quality assessment also identified long waiting times (e.g. median wait of 122 minutes for antenatal care, for a consultation lasting around 8 minutes on average [46]. However, there were wide variations by clinic: the best-performing facility had only 16% waiting over 2 hours, while the worst had half of all patients waiting over 2 hours. These data highlight significant potential in local management and organization to improve waiting times for patients.

Maintenance failures and a lack of maintenance planning exacerbate this situation. Facility maintenance is not within the control of Mpumalanga DoH and involves the Department of Public Works, Roads and Transport at provincial level. The centralised health budget makes

even minor upkeep expenses difficult to access [47]. The clinic quality assessment supports these infrastructure concerns. For example, functional internet was absent in 88% of facilities, electronic medical records were absent in 78%, and functional fans were absent in 55% [46]. Patient scores were generally high but, reflecting these issues, were low for cleanliness in particular.

Shortages of ambulances and interrupted supplies of medicine are a further critical challenge [48] reflecting the lack of autonomy on purchasing, while the Portfolio Committee 2017/18 also highlights challenges in relation to infrastructure, drug supplies, equipment, clinical staffing and referrals [34]. These issues also affect Ward-based PHC Outreach Teams work as nurses cannot reach communities to support Community Health Workers in the absence of transport [42]. Reflecting the issues described above, only 8% of clinics in Ehlanzeni met Ideal Clinic standards in 2017/18, compared to a national average of 43.5% [49].

For tracer medicines, 94% of clinics had more than 90% availability, slightly above the national average. This may reflect informal coping strategies to manage shortfalls, such as: sharing supplies across clinics; shortening the length of prescriptions; or ordering more supplies than may be warranted based on patient load owing to regularly receiving less than is ordered [46].

Finally, there are challenges relating to the quality of the District Health Information System (DHIS). While the system is widely used and tracks facility data, including child deaths, limited information is provided on cause of death and there are concerns about completeness, as well as lack of use of evidence for decision making [31];. The Portfolio Committee report for 2017/18 [34] notes that patient file management in hospitals and clinics remains a problem and that responsibility for implementation of the Health Management Information System has been moved from provincial to National DoH. A National Health Patient Registration System and web-based DHIS-2 are being rolled out [34].

## District and sub-district levels

The district and sub-district have an important role in operationalising policy, but vertical accountability can be punitive, rather than supportive, and resource constraints are commonly cited. However, good personal relations and communication can ease day to day functioning, enabling decision space to be opened. As an example, during COVID, the team observed local intersectoral action being taken in response to the crisis by district and sub-district health and other actors.

**Authority.** The district is the focus of service delivery coordination, led by the District Health Management Team. Within that team, the Primary Health Care Director is responsible for a number of programmes, including child health, which also sits under the MCWYH&N programme coordinator in terms of technical guidance. As with facility level, district and sub-district roles are clearly defined, although the MCWYH&N coordinator has responsibility for a wide range of priority programmes.

In addition, District Clinical Specialist Teams (DCST) were established in 2012 as part of the national PHC re-engineering strategy to improve quality of care, particularly for mothers and children. Each district team should include a paediatrician, for example, and teams should have autonomy to improve clinical governance. In practice, however, these teams do not have direct clinical governance mandates at district and sub-district levels but are rather advisory and supportive in terms of quality improvement, which can bring tensions [50].

Sub-districts focus on operational support to hospitals and health centres in local areas. At this level, the PHC Manager supervises implementation of priority initiatives, such as Ideal Clinics, while the MCWYH&N programme coordinator gives technical guidance and

supervision to facility staff in areas such as child health. While roles are clearly defined and understood, there is some potential for duplication (TO).

## Accountability

In the district, there is clear upward accountability through quarterly review meetings and annual reports against district health plan targets, though feedback from above tends to be focused on problems, more than identifying, understanding, supporting and enabling local leadership, supervision and innovation (such as the coping strategies around medication shortages described above). The main accountability of the MCWYH&N programme coordinator is to the District Health Management Team, rather than to technical leads at provincial level. Downward accountability links to facilities are limited, and vertical communication often depends on personal relationships. In work to date at district level, the VAPAR learning platform was seen as valuable in terms of provision of opportunities for improved vertical communication between district and provincial levels, which was identified as a major gap, as well as encouraging constructive dialogue on problems and response strategies [43].

## Capacities

Staff at district and sub-district level direct priorities for facilities but in a consultative way, for example in planning for child health campaigns. At district level, annual planning reflects divergent themes of optimism about achieving set targets while acknowledging the reality of an often under-resourced and partially dysfunctional system. Human resources challenges are highlighted in the 2018 Ehlanzeni district health plan. These include staff shortages (including support staff), high staff turnover, absenteeism, poor alignment with organisational structure and lack of outreach [47]. Hiring was decentralised to district level but then recentralised due to irregular appointments being made by district managers and CEOs. This has, however, created operational issues at district level, with long delays hiring even basic support staff (TO).

Austerity measures have been implemented annually since 2012 to improve efficiency and curb expenses. Formal and informal centralisation of procurement of goods and human resource functions has effectively limited capacity among district officials to perform duties (e.g. travelling, telephone and cell phone costs, accommodation when travelling), which is reflected in operational plans [47]. No allocation of equitable share (the funds which are send to the provinces without earmarking from national Treasury) towards goods and services was made to programmes at district and sub-district level, and with decreasing allocations at provincial level [51]. Vital equipment is reported as unavailable in some facilities, especially PHC facilities, due to budgetary constraints. Infrastructure maintenance has been failing due to centralisation of the maintenance budget, which is hard to access. Support services, such as supply chain management are slow and the competency of staff is questioned; the net effect is delays in procurement.

The sub-district has insufficient and poorly distributed community health centres (there are four when there should be 12 to serve a population of this size). Operational limitations include shortened service hours in some districts due to staff shortages, lack of transportation, poor infrastructure, lack of maintenance and ageing equipment that is insufficient in number and poorly distributed. These are all seen as contributing to a skewed utilisation of health services between PHC and secondary levels (48). Staff also report concerns over of lack of safety, both in terms of personal protection against attacks but also related to poor working environments [52].

## Provincial situation

We find that decision space at provincial level is determined by personal and relational factors, more than formal authority, accountability or capacity, which poses a risk to the performance of the health system.

## Authority

As reported at lower levels, roles and responsibilities are clearly and formally defined and widely understood. However, real and stated staffing do not always align and organograms often remain in draft for extended period and/or are outdated [34]. Nationwide, vacancies exist in programmes and posts for extended periods, and staff are widely called upon to informally fill vacant positions. Acting roles are often taken on in addition to formal roles and without delegated responsibility or remuneration [53]. The situation can contribute to authority vacuums, where staff do not feel empowered to take required decisions. At the same time, however, overstaffing of some senior management positions occurs, to the detriment of filling service delivery posts. Fluidity in roles and responsibilities is reported as a result, as well as insecurity for staff, waste and instability, with post holders' status influenced by personal networks (TO).

## Accountability

Formal structures exist for planning, budget-setting and performance targets in the Annual Performance Plan (APP) and for enforcing collective responsibility, as laid down in the Public Finance Management Act (PFMA) 1999 [54]. This sets out national standards, including the need for financial statements to be audited and made public by the national Auditor General [55]. The national Department of Planning, Monitoring and Evaluation provides cross-sectoral oversight, reporting to the President. There are a range of structures and processes in provinces: annual reviews, oversight by the Standing Committee on Public Accounts, and the Portfolio Committee on Health and Social Development under the Office of the Premier that are reported to have the power to call officials and department to account if key targets are missed [56].

Nevertheless, upward accountability to national level (or technical support from it) is seen as limited, given that health service delivery is constitutionally the responsibility of provinces, unless programmes are recipients of a national conditional grant—which only applies to HIV/ AIDS, sexually transmitted infections and TB, or issues that are in the media for some reason. Other programmes rely on annual distribution according to the provincial APP from 'equitable funds', which are allocated from the national level to the province.

There is divergence between theory and practice regarding downward linkages and accountability. While planning is designed to be bottom-up, with district plans feeding into provincial plans (59), the reverse is observed to be the case in practice. Provinces generally set priorities (including for maternal and child health) with targets divided between districts. As a result, districts lack ownership of targets and accountability is limited as performance differences are lost in aggregated provincial reporting (as long as average targets are met, individual variation is not necessarily probed) (TO). Equally, budgets are supposed to be set bottom up, according to plans, but in reality they are set top-down with managers given ceilings to work within (TO).

Moreover, programme budgets can be reallocated to different and shifting priorities (even politically driven) arising in-year, with better-connected managers often relatively protected from such variance. This undermines both performance and accountability. While there are mechanisms for enforcing collective accountability, as highlighted above, individual

accountability is less strong. The performance management and development system in the province is not functioning as intended, which means that, at an individual level, rewards for merit and sanctions for poor performance are selective (TO). Resources do exist in the system in the form of motivated staff, committed to public service, however they are often not supported. Similarly, sanctions are perceived as limited. For example, staff who are suspended pending investigation of malpractice have been reported to continue to draw salaries and benefit from housing over considerable periods, while others are moved to new positions, without the complaints being resolved [57].

The Auditor General's 2013/14 PFMA report [53] on health in the province reflects these issues, citing a lack of consequences for poor management or transgressions against the Act, poor response in addressing the root causes of poor audit outcomes, and an overall lack of key controls.

Disconnects between lower levels realities and higher level policies are widely documented in the country over sustained periods and in areas not limited to child health [58]. The VAPAR process has also identified hierarchical governance as failing to account for significant local innovation, responsiveness and resilience at lower levels. Provincial participants in the VAPAR process were positive about cooperative learning processes that enable and encourage bottom-up, appreciative and reality-based learning and exchange [42].

## Capacities

Mpumalanga had growth in PHC expenditure per capita in 2017/18, but is the third lowest spending province at R1,011 per capita; R144 below the national average [49]. Misappropriation was reported to play a role by the National Treasury, which reported fiscal risks and high accruals. Improved infrastructure, capacity enhancement, and better planning and project execution are sought to reduce under-spending, especially of conditional grants. A recent report also refers to irregular expenditure and medico-legal claims in health–litigation against the state in relation to health services, which are a growing (but unbudgeted) expenditure [34].

Annual budget increases are not aligned with population growth, extended service delivery priorities or policy directives. Priority initiatives such as Ideal Clinics, for example, have not come with additional funding. Moreover, unequal expenditure per population is reported across districts and sub-districts, highlighting the need to review budget allocations to reflect equity [51]. Budgets, which are key to enabling action, are frequently reallocated towards political priorities, such as building projects, which, without adjusted performance targets, frustrates service delivery.

Regarding staffing, appointments are often not seen to be based on merit or expertise, but rather on personal networks and years in service (TO). Trade unions also play a major role in the appointments process. These contribute to competency shortfalls, undermining managers' decision space. Organograms are highly contested and while there are estimates of required workforce, these are not matched by funding or by actual distribution of staff [51].

There have been hiring freezes [47], with brief moratoria during which there are hectic attempts to fill posts. Gaps remain, especially in previously disadvantaged areas, which is also true for facility distribution. Problems with staff morale and capacity are noted as contributing to the rise in medical litigation (mostly at hospitals and often in relation to maternity care) [34].

Although vacancy rates in critical occupations (as defined in the APP, relating to essential services) were only around 8%, there are significant challenges, given misalignment of official posts, those actually filled and those who are on payroll [30, 51]. Specialist doctors at referral hospitals were particularly lacking. 6% of employees left employment in the year covered by that report, of which a large proportion (32%) was resignations. More specifically, provincial

health management is described as existing in a state of 'crisis control', with, for example, dates for meetings set but not followed when something more urgent occurs [59].

Information flows are based on the DHIS and periodic surveys, but gaps exist, notably from the community level. In Mpumalanga specifically, a provincial health research office was established in 2016, maintaining a database of research in the province [34]. While it does not as yet actively guide or coordinate research, there are intentions to develop more input to and control over research in the province, and drawing on the VAPAR process where necessary and relevant. Mpumalanga has a relatively new university, which intends to address key health needs in future, and the province also hosts the MRC/Wits Rural Public Health and Health Transitions Research Unit (of which the HDSS is a core element)

In relation to supplies, some tenders are overseen by the Provincial Treasury, with the 'end users' in each provincial Department responsible for drafting specifications, as well as evaluating the bids received, which can be a source of distortion, with reports of mark-ups on the cost of goods and shortfalls in provision across the provincial health system, as well as poor quality goods [47].

Reflecting the issues described above, the Auditor General's PFMA report 2017/18 was qualified, with findings, which had been the case for the past five years [60]. Citing direct consequences on service organisation and delivery, it reported R310 million of irregular expenditure in 2017/18, with mutually-reinforcing capacity deficits:

> "Health, with the second largest budget, had significant findings relating to poor storage and stock management practices, staff shortages, insufficient training, and medical equipment that was not in a good working condition. These issues contributed to the poor health services in the province. The main drivers of the shortcomings were poor project management together with staff vacancies and instability."

## Discussion

Application of the decision space framework revealed system-wide patterns (Tables 2 and 3), in which lines of authority are generally well-defined in principle, however with personal networks taking on an important dimension in how stakeholders can act, and particularly when it comes to being accountable. The framework also enabled identification and description of significant informal ingenuity and capacity. The influence of political and personal factors is more clearly identified at higher system levels, across provinces and at district level, whereas at sub-district and facility level, the dominant theme is capacity, which affects all health system components. The emergent decision space is therefore characterised by personal networks and qualities, at higher levels, and on informal coping strategies between facilities and groups of staff at lower levels. Although the analysis did not focus on formally analysing changes across the period of the study, these features appear from team observations to be relatively constant over this time.

Accountability has some force collectively, in that there is real attention to meeting performance targets as set out in the APP and transparency in reporting. However, on an individual level for managers and staff, accountability is less robust and tends to be punitive and systematically screening out local innovation. For communities, direct action (often negative, in the form of protests, media stories or litigation) is perceived as more effective to enforce accountability than using the formal structures within the health system, and as documented in other sectors [61].

There are currently significant gaps between theoretical decision space and actual ability to carry out those roles (Table 3), which has implications for system performance. However,

although many challenges have been highlighted for decision space, which enables health managers and staff to respond flexibly and appropriately to local contexts, assets within the system remain rich. These include many committed staff, expertise, financial resources, well designed national policies, and willingness and ability to find informal coalitions and ways to negotiate constraints (as shown in the example of drug-sharing between clinics to avoid stock outs). Recent achievements in addressing severe acute malnutrition in the province also indicate what can be done by bringing leadership and using local data to identify priority issues and build coalitions to address them [62]. This testifies to resilience in the system and indicates how much more would be possible if current limits to decision space were addressed. The enabling of bottom-up, emergent, appreciative and reality-based learning and exchange are therefore important routes to further understand and develop decision space.

However, decision space is not likely to be sufficient to ensure good performance if mediating factors are unfavourable. A balance between the three domains of authority, accountability and capacity is important, but context also plays a very significant role. In South Africa, provinces are still struggling with the legacy of apartheid and associated socioeconomic injustices [21]. Communities were systematically exploited to provide cheap labour to benefit a white minority, explicitly economically marginalised and deprived of a decent standard of living through denial of access to well-paid work, and displaced, forced to live in 13% of the land, called Bantustans (or euphemistically homelands), most of which was agriculturally unproductive and better suited to cattle and game farming [63]. While the regime was dismantled 25 years hence, deepening social and health inequalities along the lines of race gender and economic status are widely documented [64]. Unemployment is further key contextual factor. At 35% overall, and higher among youth, it not only adds to challenges for communities in promoting health and accessing health care, but also puts pressure on the public health worker employment market [65].

Added to this is an organisational culture of low trust, related to perceptions of national level state capture, but which creates a controlling and disabling environment for staff at mid- and lower levels. Development has been hampered by patrimonialism, deployment of cadres (former anti-apartheid activists) and corruption within the ruling party [66]. 'Gatekeeper politics' ensure that those in power stay in power, and in rural areas these are predominantly elected ward councillors [66]. Consequently, those most disenfranchised are turning to struggle actions as developed during the fight against Apartheid, such as service delivery strikes.

Reflecting on how our case study complements the existing decision space literature, we make novel contributions to discussions about how sub-optimal resourcing can lead to informal coping strategies (both positive and negative) [12, 67], with health managers and staff operating as 'street level bureaucrats' having to manage in a complex environment in which their own interests and constraints play out [68]. Our findings are also consistent with studies which show the impact of resource uncertainty on narrowing decision space [68]. The difference between responsibility and financial resources to enact those responsibilities is also a common strand in the literature [12, 69], and one which we identify in the current case study, which affects not just decision space but also performance of the health system. Our case study also adds examples of the importance of infrastructure and information as enabling organisational capacities. It also highlights the importance of contextual factors in shaping decision space: even where authority, accountability and capacities may be aligned, politics and power can still disrupt, for example, with priorities and funding changing overnight at the behest of political authorities, such as the Executive Council in the province. A collaborative organisational culture and good personal relations can, on the other hand, open space at least within the local sphere and for managing problems in the short term.

Wider literature [12] suggests that bureaucratic accountability may reduce decision space in some contexts, which we also find in this setting. The perceived lack of decision-space may also have impacted on community engagement, as was documented in India [70]. In the other direction, politicised community participation may have reduced real accountability in our context.

Many studies find unclear authority as a limiting factor for decision space and accountability [12] however in this context, roles and responsibilities were relatively clear and not a major constraint. Unlike in Ghana, where incomplete political and fiscal decentralization ensured that the balance of power in the health system remained at national level [70], in Mpumalanga, there is delegation of authority through the Constitution and resources (through equalisation grants) to provincial level. However, the dynamics we document converge to squeeze decision space at lower system levels. The study is consistent with wider literature on decentralisation, finding that district teams in many settings have insufficient resources to effectively implement the health programmes they oversee [71].

The study draws on mixed sources to examine a transect through the system, but faces limitations of generalisability. However, other studies from South Africa suggest that while Mpumalanga faces some specific contextual challenges, it is broadly similar to other areas. Health district managers in Johannesburg, for example, highlighted poor leadership and planning with an under-resourced centralised approach, as well as poor communication–internally within the service and externally with the community–and poor integration of health strategies and programmes [72]. Complexity of tasks, competing demands and lack of support for front line managers and staff within a hierarchical organisational culture is also documented in other parts of the South African health system [58]. Limited capacity, inadequate operational resources and irregular monthly supervision visits have been seen to limit stewardship and poor management, with concerns about effectiveness of Ward-based PHC Outreach Teams documented in other provinces [73–75]. Further, Coovadia et al. highlight widespread challenges of gaps in competence and lack of effective leadership, stewardship and personal accountability, rooted in historical legacies of colonialism and apartheid. A recent study also found growing corruption in the health sector in South Africa, with the largest number of reports focussing on the provincial level [76].

The recent South African Quality of Care Commission's first recommendation focuses on ethical and effective leadership, while its second is on strengthening community structures for engagement and accountability [77]. Research conducted in Mpumalanga, Gauteng and Western Cape also highlighted the lack of accountability to patients, and tendency to focus on upward accountability instead, which supports abusive relationships with patients but is also driven in part by lack of support for staff [78]. While many quality improvement initiatives are focused at facility level, these will struggle to be effective and be sustained without more meso and macro level changes [79].

Some express concern that the compliance culture in the public sector as a whole–deepened by recent anti-corruption measures—is crowding out innovation, developmental, non-hierarchical and cross-departmental approaches and responsiveness to users:

> *"Protests happen every day, but officials worry more about what the Auditor-General will say, or whether politicians will throw them under the bus, than what the people think of them* [80]*"*

Debate concerns whether new political windows of opportunity are opening in South Africa currently, which can combat corruption, challenge the culture of cover up and open up

decision spaces for committed and skilled managers at all levels of the health (and wider public) system.

While these factors are amenable to change, it is important to acknowledge the deep structural influences from social, historical and health systems contexts and organisational cultures. Our analysis above indicates that it will take time and commitment to motivate staff and provide meaningful, distributed leadership. Support for informal leadership development strategies may also be an important element in building capacity at lower system levels to expand and use decision space [81], along with a greater focus on system 'software', such as building trusting relationships, improved communication and dialogue skills. Better communication and direct contact between government officials and citizens are also key [82]. A review of the VAPAR process to date suggests that willingness and commitment for cooperative reflection and action exists within the local health system, evidenced in both sustained engagement and participation in the process and formalised partnerships for health systems [41]. In other provinces, a 'war-room' approach to tackling priority child health challenges at ward level, involving multiple stakeholder groups, was reported to be effective [43].

This cooperative action, including across sectors, has been in evidence in the recent effort to tackle COVID in the province, as evidenced by the formation of municipal, district and provincial level joint operations committees, to which all departments report at least once a week. The national but focused nature of this crisis appears to have enabled multisectoral collaboration. However, that does not necessarily imply local decision-space: community health workers were, for example, given new roles in COVID messaging–away from their previous focus on areas such as child growth monitoring and defaulter tracing–without local consultation with the CHWs themselves [83].

## Study limitations

A major limitation of the paper is that the examination of decision space was not planned in advance but arose from an observation of its importance in influencing how evidence co-created by VAPAR partners could be put into use. The analysis was therefore conducted retrospectively and limited by available data. As a result, the paper is not able to assess decision space (in relation to different health system functions) or its impact on performance. However, we are able to observe inter-relationships between components which influence decision-space and comment on how constraints to these might be mitigated and opportunities taken to improve them.

## Conclusions

In the context of a participatory action research programme, we set out to establish what the margins for reform and for coproducing and responding to evidence were across the health system in a rural South African province. We used a decision space framework, which focused on the relationships between authority, accountability and capacities in shaping the power to decide and act of local health managers and staff.

We find that capacity exists in the system, but accessing it is frequently made difficult due to a number of intervening factors. While lines of authority are generally well-defined in theory, personal networks take on an important dimension in practice in how stakeholders can act. This is expressed through ingenuity and a range of informal coping strategies built on local relationships. There are constraints in terms of limited formal external accountability to communities, and internal accountability which is weak in places for individuals and focused more on meeting higher level performance targets and less on enabling local leadership. More generally, political and personal factors are clearly identified at higher levels of the system

whereas at sub-district and facility levels, the dominant theme was capacity, affecting all health system components.

Decision space is particularly relevant for child health, given the acute and life-threatening nature of child health conditions. Such space as exists is fragile, and can be lost due to political, organisational and other shocks, such as COVID. Creating spaces to support more constructive relationships and dialogue across system levels emerges as important, as well as reinforcing horizontal, peer-to-peer networks to problem solve at local level, and developing capacity of link-agents such as community health workers to increase community accountability. In this scenario, the main resources in the system, which are human, technical and financial, can be energised to realise potential more fully.

## Supporting information

**S1 Checklist.**
(DOCX)

## Acknowledgments

We would like to thank all community and health system participants whose insights and experiences informed this article. We thank ReBUILD for support with the publication costs.

## Author Contributions

**Conceptualization:** Sophie Witter, Maria van der Merwe, Rhian Twine, Stephen M. Tollman, Lucia D'Ambruoso.

**Data curation:** Sophie Witter, Maria van der Merwe, Denny Mabetha, Jennifer Hove, Lucia D'Ambruoso.

**Formal analysis:** Sophie Witter, Maria van der Merwe, Denny Mabetha, Jennifer Hove, Stephen M. Tollman, Lucia D'Ambruoso.

**Funding acquisition:** Sophie Witter, Maria van der Merwe, Rhian Twine, Stephen M. Tollman, Lucia D'Ambruoso.

**Investigation:** Sophie Witter, Maria van der Merwe, Lucia D'Ambruoso.

**Methodology:** Sophie Witter, Maria van der Merwe, Lucia D'Ambruoso.

**Project administration:** Sophie Witter, Maria van der Merwe, Rhian Twine, Denny Mabetha, Jennifer Hove, Lucia D'Ambruoso.

**Resources:** Maria van der Merwe, Stephen M. Tollman, Lucia D'Ambruoso.

**Writing – original draft:** Sophie Witter.

**Writing – review & editing:** Sophie Witter, Maria van der Merwe, Rhian Twine, Denny Mabetha, Jennifer Hove, Stephen M. Tollman, Lucia D'Ambruoso.

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
