## [Decision Letter · Decision Letter 0]

8 Nov 2023

PONE-D-22-22471Opening decision spaces: a case study on the opportunities and constraints in the public health sector of Mpumalanga Province, South AfricaPLOS ONE

Dear Dr. Lucia D'Ambruoso,

Thank you for submitting your manuscript to PLOS ONE. After careful consideration, we feel that it has merit but does not fully meet PLOS ONE’s publication criteria as it currently stands. Therefore, we invite you to submit a revised version of the manuscript that addresses the points raised during the review process.

ACADEMIC EDITOR:The authors are to be congratulated for this pertinent body of work. However, there are a few minor changes that will need to be addressed.In the abstract, Line 18. Locally informed and locally relevant. Locally doesn't need to be repeated.Abstract results. Although capacity exists. Capacity for what. Add capacity for decision making.Introduction, line 61. VAPAR is mentioned for the first time. Put a reference.Data sources, line 200. Spell out VA and PAR before abbreviating. Go through the entire manuscript and fully spell out all abbreviations used for the first time e.g. Line 226 NGOs.Study limitations should come after the discussion.Footnotes are not permitted. If your manuscript contains footnotes, move the information into the main text or the reference list, depending on the content.Provide full references for references 2, 9, 17, 19, 56 and 57

We look forward to receiving your revised manuscript.

Kind regards,

Ada Aghaji

Academic Editor

PLOS ONE

Journal Requirements:

Reviewers' comments:

Reviewer's Responses to Questions

**Comments to the Author**

1. Is the manuscript technically sound, and do the data support the conclusions?

Reviewer #1: Yes

2. Has the statistical analysis been performed appropriately and rigorously? 

Reviewer #1: N/A

3. Have the authors made all data underlying the findings in their manuscript fully available?

Reviewer #1: No

4. Is the manuscript presented in an intelligible fashion and written in standard English?

Reviewer #1: Yes

5. Review Comments to the Author

Reviewer #1: Opening decision spaces: a case study on the opportunities and constraints in the public health sector of Mpumalanga Province, South Africa

Reveiwer comments

General comments

This is a well written paper on a very important issue for federating low and middle income countries where the subnational level are mandated to implement health policies. Below are specific comments and suggestions for minor revisions

Specific comments

Abstract

Line 25..remove the, and leave as ....three core domains

Key words... Iw ould have expected to see ‘ Decentralisation’ as a key word, since it’s the principle at the core of subnational decision making

Introduction

I suggest that it nay be necessary to further explain ( in a few swntences), the model of decentralization ( deconcentration, devolution etc.) that is in practice as this may have resonance for the findings. And sometimes actors, in practice, deviate from the model prescribed by their constitution, especially in LMICs

Methods

I commend the detailed writing in this section and suggest a few revisions, as follows:

Lines 160-165

It will be good to rank the socio-economic status of Mpumalaga among the other provinces, for the wider readership to fully appreciate the context, especially as Capacity ( resource... financial etc.) is one of your key study domains. The impact of resource capacity on the decision space may well differ in a better resourced provice, like the Western Cape, for instance.

Line 195....... remove one ‘from’

Line 202...Edit sentence

Line 211..... Though referenced, I think a sentence on the purpose of selecting these particular three villages can be included here, without the reader going to seek out the Reference.

Results

Findings have also been reported in a detailed manner

Given the long study period, (2015-2020), albeit retrospective, were there fluctuations observed in the decision spaces at the various levels and what factors (political, economic, change in actors) may have accounted for these?

Discussion

You now bring in Decentralisation literature here, which is apt, but not adequately introduced at the beginning.

I also wonder if some street level bureaucracy literature can be used to explain the facility staff coping mechanism (positive or negative).

6. PLOS authors have the option to publish the peer review history of their article (what does this mean?). If published, this will include your full peer review and any attached files.

Reviewer #1: No

---

## [Author Response · Author response to Decision Letter 0]

15 Dec 2023

Many thanks for your very constructive review. Please find our responses in blue below.

Reviewer #1: Opening decision spaces: a case study on the opportunities and constraints in the public health sector of Mpumalanga Province, South Africa

General comments

This is a well written paper on a very important issue for federating low and middle income countries where the subnational level are mandated to implement health policies. Below are specific comments and suggestions for minor revisions

Specific comments

Abstract

Line 25..remove the, and leave as ....three core domains

This has been done.

Key words... I would have expected to see ‘ Decentralisation’ as a key word, since it’s the principle at the core of subnational decision making

Thanks for noting this omission, this was related to the fact that the health system was decentralised some time ago in South Africa (so this is not an ongoing process, as it is in many settings where studies are conducted). However, we agree that it is relevant here and have now added it.

Introduction

I suggest that it may be necessary to further explain ( in a few sentences), the model of decentralization ( deconcentration, devolution etc.) that is in practice as this may have resonance for the findings. And sometimes actors, in practice, deviate from the model prescribed by their constitution, especially in LMICs

This is an administrative decentralisation (not decongestion or devolution), with functions mandated to the provincial level in particular. This is explained in the first paragraph but we have added this qualifier. Later in the paper some of the challenges of resource flows still emanating from the central level are elaborated.

Methods

I commend the detailed writing in this section and suggest a few revisions, as follows:

Lines 160-165

It will be good to rank the socio-economic status of Mpumalaga among the other provinces, for the wider readership to fully appreciate the context, especially as Capacity ( resource... financial etc.) is one of your key study domains. The impact of resource capacity on the decision space may well differ in a better resourced provice, like the Western Cape, for instance.

We have added a comment on the GDP per capita, though note that there are equalisation grants so resources for health care are not solely dependent on the local economy.

Line 195....... remove one ‘from’

Done

Line 202...Edit sentence

Done, thank you for picking up this typo.

Line 211..... Though referenced, I think a sentence on the purpose of selecting these particular three villages can be included here, without the reader going to seek out the Reference.

We have added a short description on how these villages were selected.

Results

Findings have also been reported in a detailed manner

Given the long study period, (2015-2020), albeit retrospective, were there fluctuations observed in the decision spaces at the various levels and what factors (political, economic, change in actors) may have accounted for these?

We did not analyse trends over the period formally, and so cannot add this to findings but we have added a comment in the discussion that these features appear to remain valid, i.e. have not significantly fluctuated over the period, to our knowledge.

Discussion

You now bring in Decentralisation literature here, which is apt, but not adequately introduced at the beginning.

I also wonder if some street level bureaucracy literature can be used to explain the facility staff coping mechanism (positive or negative).

Thank you, that is indeed a relevant concept to draw on and we have added this to the discussion, though we were mindful of not adding too much length to what is already a long paper.

ACADEMIC EDITOR:

The authors are to be congratulated for this pertinent body of work. However, there are a few minor changes that will need to be addressed.

In the abstract, Line 18. Locally informed and locally relevant. Locally doesn't need to be repeated.

Deleted, thanks

Abstract results. Although capacity exists. Capacity for what. Add capacity for decision making.

Added, thanks

Introduction, line 61. VAPAR is mentioned for the first time. Put a reference.

We have added the programme website as a reference here

Data sources, line 200. Spell out VA and PAR before abbreviating. Go through the entire manuscript and fully spell out all abbreviations used for the first time e.g. Line 226 NGOs.

We have checked these and added/corrected, also to the abbreviation list

Study limitations should come after the discussion.

We have moved this sub-section down

Footnotes are not permitted. If your manuscript contains footnotes, move the information into the main text or the reference list, depending on the content.

We have turned the footnotes into references

Provide full references for references 2, 9, 17, 19, 56 and 57

We have added details to these references as required

---

## [Decision Letter · Decision Letter 1]

29 Jan 2024

PONE-D-22-22471R1Opening decision spaces: a case study on the opportunities and constraints in the public health sector of Mpumalanga Province, South AfricaPLOS ONE

Dear Dr. D'Ambruoso,

Thank you for submitting your manuscript to PLOS ONE. After careful consideration, we feel that it has merit but does not fully meet PLOS ONE’s publication criteria as it currently stands. Therefore, we invite you to submit a revised version of the manuscript that addresses the points raised during the review process.

Please address comments raised by Reviewer 4.

We look forward to receiving your revised manuscript.

Kind regards,

Sogo France Matlala, PhD

Academic Editor

PLOS ONE

Reviewers' comments:

Reviewer's Responses to Questions

**Comments to the Author**

1. If the authors have adequately addressed your comments raised in a previous round of review and you feel that this manuscript is now acceptable for publication, you may indicate that here to bypass the “Comments to the Author” section, enter your conflict of interest statement in the “Confidential to Editor” section, and submit your "Accept" recommendation.

Reviewer #2: (No Response)

Reviewer #3: All comments have been addressed

Reviewer #4: (No Response)

2. Is the manuscript technically sound, and do the data support the conclusions?

Reviewer #2: Partly

Reviewer #3: Yes

Reviewer #4: Partly

3. Has the statistical analysis been performed appropriately and rigorously? 

Reviewer #2: N/A

Reviewer #3: N/A

Reviewer #4: N/A

4. Have the authors made all data underlying the findings in their manuscript fully available?

Reviewer #2: Yes

Reviewer #3: Yes

Reviewer #4: No

5. Is the manuscript presented in an intelligible fashion and written in standard English?

Reviewer #2: No

Reviewer #3: Yes

Reviewer #4: Yes

6. Review Comments to the Author

Reviewer #2: The main issue is that the manuscript involve two things which could have led to two separate manuscripts: (1) application of the concept framework in decison making space, and also (2) assessment of decision making space using participatory approach.

As such this has impacted your background, methodology, results and discussions.

generate two manuscript from this submission

Reviewer #3: The manuscript is recommended for publication. The authors to justify as to why the three villages were selcted above all other available villages. Also, attend to the referencing of government documents as per given example in the manuscript.

Reviewer #4: Thank you for submitting this interesting paper to review. You use an interesting methodology using PAR to unpack the decision-making space issue. I have a number of overarching comments and questions as well as specific ones related to the text.

You highlight the importance of the decentralised health system in South Africa – which is administratively decentralised to the district level. You introductory paragraph remarks that the health system is operationalised by the districts in line (line 57-8) with standards and set targets for population level indicators. Who does the target setting? National? Provinces? To what extent is this determined at a district level in Mpumalanga? What is the target setting process/consultation process? How does local evidence impact on target setting and therefore resource allocation? Unpacking some of this could assist with understanding the gaps in ownership of decisions, and the architecture of the decision space. You conclude the article with a discussion about making spaces. Not much mention is made about the absence of these in the current district functioning.

In addition, the issue of adequate information for astute decision making is not explicitly highlighted. Poor information management and reporting of metrics that span the whole province with little drilling down to district or facility level surely constrains decision making and resource allocation, particularly in a resource constrained environment.

For some specific comments:

1. Line 91 mentions resource capacities – are these technical (is this where data comes in as well as the ability to make sense of data – i.e. information)? You mention the provincial M&E office but do not draw this into the research or discussion. This is important as poor access to valid information informs resource decision-making.

2. You point to the implications of inadequate resource capacities – with reverting to informal decision-making strategies. Do you mean that for example if there is little information available informal decision-making emerges?

3. You mention (line 109-110) that leadership should enable front line staff to make decisions about delivery to meet needs. However, your phrasing indicates that you see in the SA context that this is for service delivery that fulfils policy mandates.

4. The accountability paragraph reviews horizontal and vertical decision making. Community accountability through structures are the only ones mentioned in horizontal accountability. However there could be on-line forums that promote discussion and problem solving that are not decision-making forums that are now used in some provincial health departments, e,g, the Western Cape. These are designed for district and provincial managers to touch sides about mutual problems, highlighting conundrums, how they have managed them, promoting innovation and spreading good practice. This has engendered more trusting relationships and less of compliance culture. Your consideration of opportunities brought about by digital resources would be useful.

5. Line 143: What do you mean by “affirmative student recruitment”? Is this purposefully recruiting local students to work in a rural province? Other?

6. Line 144: your use of ‘public health workforce”. Is this a public sector health workforce”, or staff with public health skills (epidemiology, management etc)?

7. Line 163 – suggest you replace ‘it’ with ‘this’.

8. In your methods you mention that the data was obtained through team observations (TO). However it is not clear if the opportunities for review of data, and mutual support by managers at a provincial, district or local level were probed by the team.

9. Line 321. The mention of ‘scripts’: Is this scripts to be used to conduct consultations or decision tree/algorithmic/ protocols as to what the indications are for various conditions e.g. the PACK guidelines that the Knowledge Translation Unit design.

10. The initiatives (line 323) that ‘lacking local relevance” is interesting, and could be unpacked in more detail. Are there diseases/conditions that are not covered or management protocols that are not possible or relevant?

11. You make mention that the relationships between the hospitals and CHCs are no longer direct. How do the Paediatricians instituted by the WBOTS function ito supervision? How do they relate to hospital clinicians? In line 403 you mention that there ‘may’ be confusion over mandates? Is there confusion? Was this not probed? This is key to an exploration of decision spaces. If there is no WBOTs or discipline specialists are there any other clinical supervision plans made? The PHC supervisors, are they clinically trained experienced clinical nurse practitioners?

12. You give figures for anticipated attrition (line 349-350). Was there a correlation between the clinics with high levels of staff dissatisfaction and high proportions of staff intending to leave?

13. Line 355 – “area was a homeland” – this needs more contextual explanation, as only those familiar with apartheid implementation would know anything about this. You only raise homelands in the discussion. I think this should come earlier.

14. The link between capacity, decision spaces and long waiting times needs to be made clearer. Do clinic managers have the mandate, information and tools to make decisions about implementing booking systems for patient consultations that would reduce patient waiting times ?

15. Why do you think the proportion of clinics have met Ideal Clinic standards, compared to the national average, are so low (line375-376). Surely all provinces have the same constraints and consequently procurement constraints cannot be the problem.

16. The report on deficiencies in the use of data to determine causes of death via the DHIS merits further discussion. Are the underlying causes of hospital deaths not recorded in the hospital information system? In addition, are there not M&M meetings to discuss hospital deaths? Also unexplained deaths should be investigated by the Forensic pathology services. Surly there are some avenues open to obtain some sense of leading causes of death in children in the district?

17. Line 397. It would be helpful to have more detail about the PHC director who “sits under the MCWYH&N programme coordinator”. Is the MCWYH&N person a director as well? Are they located in the provincial office? The job description for accountabilities would be useful. While this may seem pernickety, I think this is pertinent as decision making, accountability and lines of authority are the subject of the paper.

18. The perspectives of the district authority about the roles and reporting lines of the DCST would be useful as clinical governance is not independent of service targets. Perhaps the DCSTs were imposed on the district who had little control over their activities?

19. Lines 412-422: The district accountability upward and downward accountability mechanisms seem to be in the form of written reports? Besides the VAPAR forum (a district level meeting?), are there any discussion forums within the district – such as monthly district / sub-district meetings where issues can be raised? What is the level of understanding of information derived from data that is reported which may lend itself to identifying gaps in ?training ?procurement ?staffing.

20. Line 427: You assert that the system is dysfunctional. That is a sweeping comment. It would be useful to have better precision here, e.g. people management/HR problems with the detail. Was the recentralisation to the provincial level? How were the irregularities managed?

21. Line 434-438: I think there should be a full-stop Line 435 “duties”. The “however” sentence is grammatically problematic.

22. The “equitable share” comment should include what was funded? Only the HR component?

23. The lack of ownership by districts of provincial targets points to a top-down imposition of targets. This may point to inadequate forums for decision making and/or a provincial office that is resource constrained or has inadequate management practice.

24. The people management systems at an individual level are highlighted as problematic. How do performance appraisals (6 monthly and annual) work. Surely each individual is appraised by their line manager at least annually? Or is this not currently functioning?

25. You point out that those suspended continue to draw salaries, with an implied criticism. Do you think they should not? Surely the issue here is the quicker resolution of such cases? The onus of proof should be on the complainant’s strong case.

26. You point to the damning Auditor General’s 2013/14 report (line 501). This is long out of date. I see that you do quote the 2017/18 report (line552). Can you identify if there have been improvements/changes in the AG’s assessment. It may be useful to understand the trajectory. I see that in the Discussion, you say that you did not analyse changes over time. How did you assess that “these features appear to be relatively constant over time” (line 577-8)?

27. Line 545 suggest that “intends” should be ‘intend”. Does the new University host the Wits/MRC Unit currently? How is it intended that this will work?

28. The ‘end users’ referred to (line 548): are these district or hospital or facility managers.

29. Table 3: This is a useful summary of findings. The comment on reporting – quantitative data as a limitation is only compared to social factors. Are there not other quantitative data that could be elicited that are not, e.g. retention in care or transfers between levels of care for a myriad of health conditions?

30. In the discussion, you raise that political and personal factors are identified across provinces. You do not discuss this in the results, and it is unclear where there is evidence for this. Relevant in this context?

31. You raise the enabling of learning and exchange that are critical to the development of decision-space. What kinds of decision spaces would facilitate this? Is it a matter of capacity such as developing leadership skills among middle and senior managers?

32. The contextual factors raised in the discussion – apartheid legacy issues, could be raised in the introduction which would give the reader more background and inform their understanding of the research findings.

33. It is unclear what is meant by (line 610) “public health employment market”. Do you mean that the public sector is a key employer for aspirant job seekers?

34. Line 611-12 could be clearer: “perceptions of higher-level distortions and state capture” – be more explicit - are you talking about national government? About political interference in the administration? About external organisations wishing to take control of state assets? The assertions are too sweeping with little detail. Whose perceptions? I think rephrase.

35. The explanation of Gatekeeper politics related to ward counsellors point to local government, and local government is NOT the custodian of health service delivery. It is unclear why this is raised here.

36. Line 623. “The difference between responsibility and financial resources …” is unclear Are you saying there is a mismatch, and what are the implications?

37. You seem to indicate that authority, accountability and capacities are aligned (line 628 & line637). Is this the case? What do you mean by aligned? Are the WBOTs aligned with district functions? And are there decision spaces such as regular district or provincial meetings that are functional?

38. You point out that there are insufficient management skills (line 643),. Is this not a deficit at a ‘capacity’ level? Does this not indicate poor alignment between capacity, authority and accountability?

39. The conclusion summarises the findings which is helpful. However could you be clearer as to what you are saying about lines of authority. Although they may be well defined are they functional? And if they were functional would there be a need for local ingenuity?

40. Your concluding commentary – the creation of spaces for constructive dialogue and decision making – is ‘spot on’ and more could be unpacked about this in the course of the results and discussion.

7. PLOS authors have the option to publish the peer review history of their article (what does this mean?). If published, this will include your full peer review and any attached files.

Reviewer #2: No

Reviewer #3: No

Reviewer #4: No

---

## [Author Response · Author response to Decision Letter 1]

7 May 2024

Thank you for these additional comments. We provide responses in blue below.

Reviewer #4: Thank you for submitting this interesting paper to review. You use an interesting methodology using PAR to unpack the decision-making space issue. I have a number of overarching comments and questions as well as specific ones related to the text.

You highlight the importance of the decentralised health system in South Africa – which is administratively decentralised to the district level. You introductory paragraph remarks that the health system is operationalised by the districts in line (line 57-8) with standards and set targets for population level indicators. Who does the target setting? National? Provinces? To what extent is this determined at a district level in Mpumalanga? What is the target setting process/consultation process? How does local evidence impact on target setting and therefore resource allocation? Unpacking some of this could assist with understanding the gaps in ownership of decisions, and the architecture of the decision space. You conclude the article with a discussion about making spaces. Not much mention is made about the absence of these in the current district functioning.

Our focus is on the province (and the levels within it, including the district) as health is devolved to the provinces in South Africa. We have added some clarifications to this first paragraph to make that clearer. The national level sets the policy framework and norms but the provinces set specific targets for service coverage etc internally. They consult in this process but the bottom-up planning has its limitations, as we explore in the paper (e.g. see lines 481-88 on the provincial target setting approach). The districts are responsible primarily for service delivery.

In addition, the issue of adequate information for astute decision making is not explicitly highlighted. Poor information management and reporting of metrics that span the whole province with little drilling down to district or facility level surely constrains decision making and resource allocation, particularly in a resource constrained environment.

Yes, agreed that information and its use is key, which is why we address this (as one of the system blocks) in the capacity sections (e.g. 381-87 and 540-47). The information system does extend to district and facility levels but there are gaps, as we highlight, especially in relation to community data, and the largely top-down planning limits the incentive to use local data for priority-setting.

For some specific comments:

1. Line 91 mentions resource capacities – are these technical (is this where data comes in as well as the ability to make sense of data – i.e. information)? You mention the provincial M&E office but do not draw this into the research or discussion. This is important as poor access to valid information informs resource decision-making.

We define capacities in this paragraph as follows: ‘Capacity is what enables the organisation to function (7), and includes administrative, technical, organisational, financial and human resources’. So it includes technical. We have added information/data here now (it was implied but not explicit). We do discuss this throughout the paper but as we are covering multiple levels, domains and system blocks, we have had to keep each area succinct (the paper is already long).

2. You point to the implications of inadequate resource capacities – with reverting to informal decision-making strategies. Do you mean that for example if there is little information available informal decision-making emerges?

Exactly. We are highlighting the interdependence of the three aspects of our framework in this section – so in the example of information systems, local leaders need resources, such as valid data, but also a clear authority to use it for decision making, and also being held accountable if they don’t do so. Informal decision making can relate to lack of clear authority, as well as lack of capacity, for example. We have done a small edit here to make the text easier to follow.

3. You mention (line 109-110) that leadership should enable front line staff to make decisions about delivery to meet needs. However, your phrasing indicates that you see in the SA context that this is for service delivery that fulfils policy mandates.

Indeed, at this point (definitions and framework section) we are highlighting this tension but we go on to explore it more fully in the rest of the article.

4. The accountability paragraph reviews horizontal and vertical decision making. Community accountability through structures are the only ones mentioned in horizontal accountability. However there could be on-line forums that promote discussion and problem solving that are not decision-making forums that are now used in some provincial health departments, e,g, the Western Cape. These are designed for district and provincial managers to touch sides about mutual problems, highlighting conundrums, how they have managed them, promoting innovation and spreading good practice. This has engendered more trusting relationships and less of compliance culture. Your consideration of opportunities brought about by digital resources would be useful.

Fora which are used for learning and peer-to-peer support and sharing are crucial for learning health systems but we would probably not site these within accountability, as they serve a different purpose. 

5. Line 143: What do you mean by “affirmative student recruitment”? Is this purposefully recruiting local students to work in a rural province? Other?

We have added a clarification here, this is about giving priority to students from historically disadvantaged backgrounds. 

6. Line 144: your use of ‘public health workforce”. Is this a public sector health workforce”, or staff with public health skills (epidemiology, management etc)?

Always ambiguous in English! We have added public sector to make this clearer.

7. Line 163 – suggest you replace ‘it’ with ‘this’.

Done

8. In your methods you mention that the data was obtained through team observations (TO). However it is not clear if the opportunities for review of data, and mutual support by managers at a provincial, district or local level were probed by the team.

All areas of health system blocks were probed – we have added this more explicitly to line 240 in the methods. However, as stated, we have had to condense the descriptions by health system functional area (governance, financing, information systems, supplies, HRH etc) across each level of the provincial health system to make the article manageable in length.

9. Line 321. The mention of ‘scripts’: Is this scripts to be used to conduct consultations or decision tree/algorithmic/ protocols as to what the indications are for various conditions e.g. the PACK guidelines that the Knowledge Translation Unit design.

We have edited the sentence as scripts may give the wrong impression; these are programme guidance documents for each main service delivery area.

10. The initiatives (line 323) that ‘lacking local relevance” is interesting, and could be unpacked in more detail. Are there diseases/conditions that are not covered or management protocols that are not possible or relevant?

We try to elaborate briefly but these can include mandates that do not fit to local resources, or priorities which do not reflect local conditions.

11. You make mention that the relationships between the hospitals and CHCs are no longer direct. How do the Paediatricians instituted by the WBOTS function ito supervision? How do they relate to hospital clinicians? In line 403 you mention that there ‘may’ be confusion over mandates? Is there confusion? Was this not probed? This is key to an exploration of decision spaces. If there is no WBOTs or discipline specialists are there any other clinical supervision plans made? The PHC supervisors, are they clinically trained experienced clinical nurse practitioners?

This was probed but the situation on the ground is marked by some ambiguity. The paediatricians are supportive/advisory but do not have direct supervisory roles. We have edited this section to make it clearer. The PHC supervisors are trained nurses, and we have added this specification.

12. You give figures for anticipated attrition (line 349-350). Was there a correlation between the clinics with high levels of staff dissatisfaction and high proportions of staff intending to leave?

This was an independent study so we are only able to highlight relevant findings that they report; we are not sure if there is a correlation of this kind, apologies.

13. Line 355 – “area was a homeland” – this needs more contextual explanation, as only those familiar with apartheid implementation would know anything about this. You only raise homelands in the discussion. I think this should come earlier.

We have added a small explainer here.

14. The link between capacity, decision spaces and long waiting times needs to be made clearer. Do clinic managers have the mandate, information and tools to make decisions about implementing booking systems for patient consultations that would reduce patient waiting times ?

The variable performance across clinics in the same area suggests that they can ameliorate waiting times to some degree; however, we are reporting findings from another study here so do not have full details on the enablers and constraints.

15. Why do you think the proportion of clinics have met Ideal Clinic standards, compared to the national average, are so low (line375-376). Surely all provinces have the same constraints and consequently procurement constraints cannot be the problem.

The whole premise of the article is that all provinces do not face the same challenges to decision space. There is considerable variation in this decentralised system in relation to resources, governance, capacity and performance. Supply systems are not the same for all provinces. Nor is probity etc. 

16. The report on deficiencies in the use of data to determine causes of death via the DHIS merits further discussion. Are the underlying causes of hospital deaths not recorded in the hospital information system? In addition, are there not M&M meetings to discuss hospital deaths? Also unexplained deaths should be investigated by the Forensic pathology services. Surly there are some avenues open to obtain some sense of leading causes of death in children in the district?

The main challenge with DHIS mortality data concerns data quality. In Mpumalanga, child deaths are separately recorded and reviewed by a dedicated official within the Child Problem Identification Programme (PIP), with a parallel database reflecting the causes of death, which differs notably from what has been recorded on the DHIS. 

17. Line 397. It would be helpful to have more detail about the PHC director who “sits under the MCWYH&N programme coordinator”. Is the MCWYH&N person a director as well? Are they located in the provincial office? The job description for accountabilities would be useful. While this may seem pernickety, I think this is pertinent as decision making, accountability and lines of authority are the subject of the paper.

There is an MCWYH&N director at provincial level but also at district (which is the level being discussed in this section). They provide technical support to many of the important health programmes, but managerial control sits under the PHC director. We have again tried to clarify briefly.

18. The perspectives of the district authority about the roles and reporting lines of the DCST would be useful as clinical governance is not independent of service targets. Perhaps the DCSTs were imposed on the district who had little control over their activities?

Indeed, there has been some tension around their role. We have indicated that here more explicitly now.

19. Lines 412-422: The district accountability upward and downward accountability mechanisms seem to be in the form of written reports? Besides the VAPAR forum (a district level meeting?), are there any discussion forums within the district – such as monthly district / sub-district meetings where issues can be raised? What is the level of understanding of information derived from data that is reported which may lend itself to identifying gaps in ?training ?procurement ?staffing.

There are regular meetings where issues are raised, we have made that clearer now, however, the interaction between the levels can be poor, as we outline, with a focus on fault finding more than supporting solutions. Some of these issues are about organisational culture, more than data.

20. Line 427: You assert that the system is dysfunctional. That is a sweeping comment. It would be useful to have better precision here, e.g. people management/HR problems with the detail. Was the recentralisation to the provincial level? How were the irregularities managed?

We do give details here of some of the key problematic areas but agree that it is not fully dysfunctional (now edited to partially). The HRH area in particular is challenging and has been subject to a lot of contestation, including political and patronage-based.

21. Line 434-438: I think there should be a full-stop Line 435 “duties”. The “however” sentence is grammatically problematic.

Deleted, thanks

22. The “equitable share” comment should include what was funded? Only the HR component?

The equitable share is un-earmarked funds to provinces; how they use it internally is up to them and we are highlighting that they have not allocated it to health services here. We have added a short explainer.

23. The lack of ownership by districts of provincial targets points to a top-down imposition of targets. This may point to inadequate forums for decision making and/or a provincial office that is resource constrained or has inadequate management practice.

Agreed, and we try to unpack these issues throughout, which include more complex problems like poor organisational culture and top leadership.

24. The people management systems at an individual level are highlighted as problematic. How do performance appraisals (6 monthly and annual) work. Surely each individual is appraised by their line manager at least annually? Or is this not currently functioning?

Formal systems may exist but personal networks appear to be key to getting and retaining posts at provincial level, as we highlight (current line 469).

25. You point out that those suspended continue to draw salaries, with an implied criticism. Do you think they should not? Surely the issue here is the quicker resolution of such cases? The onus of proof should be on the complainant’s strong case.

The issue is when complaints are parked and not resolved, leaving people in stasis, or simply moved to avoid the issue. We add a clarifier here.

26. You point to the damning Auditor General’s 2013/14 report (line 501). This is long out of date. I see that you do quote the 2017/18 report (line552). Can you identify if there have been improvements/changes in the AG’s assessment. It may be useful to understand the trajectory. I see that in the Discussion, you say that you did not analyse changes over time. How did you assess that “these features appear to be relatively constant over time” (line 577-8)?

The 2017/18 report is on a different topic (public financial management, not health specifically). In relation to change, we have added an explainer to the discussion: we did not formally analyse data on change over time, but from working in the system we observe this (now clearly stated).

27. Line 545 suggest that “intends” should be ‘intend”. Does the new University host the Wits/MRC Unit currently? How is it intended that this will work?

Corrected and clarified: the province hosts the unit (not the new university).

28. The ‘end users’ referred to (line 548): are these district or hospital or facility managers.

These are provincial DoH managers. Clarified.

29. Table 3: This is a useful summary of findings. The comment on reporting – quantitative data as a limitation is only compared to social factors. Are there not other quantitative data that could be elicited that are not, e.g. retention in care or transfers between levels of care for a myriad of health conditions?

Thanks, we have a

---

## [Decision Letter · Decision Letter 2]

20 May 2024

Opening decision spaces: a case study on the opportunities and constraints in the public health sector of Mpumalanga Province, South Africa

PONE-D-22-22471R2

Dear Dr. Lucia D'Ambruoso

We’re pleased to inform you that your manuscript has been judged scientifically suitable for publication and will be formally accepted for publication once it meets all outstanding technical requirements.

Kind regards,

Sogo France Matlala, PhD

Academic Editor

PLOS ONE

Additional Editor Comments (optional):

Reviewers' comments:

Reviewer's Responses to Questions

**Comments to the Author**

1. If the authors have adequately addressed your comments raised in a previous round of review and you feel that this manuscript is now acceptable for publication, you may indicate that here to bypass the “Comments to the Author” section, enter your conflict of interest statement in the “Confidential to Editor” section, and submit your "Accept" recommendation.

Reviewer #4: All comments have been addressed

2. Is the manuscript technically sound, and do the data support the conclusions?

Reviewer #4: Yes

3. Has the statistical analysis been performed appropriately and rigorously? 

Reviewer #4: N/A

4. Have the authors made all data underlying the findings in their manuscript fully available?

Reviewer #4: No

5. Is the manuscript presented in an intelligible fashion and written in standard English?

Reviewer #4: Yes

6. Review Comments to the Author

Reviewer #4: Thank you for addressing comments so comprehensively. The changes to the document have dealt with the concerns.

Congratulations for producing such an interesting article.

7. PLOS authors have the option to publish the peer review history of their article (what does this mean?). If published, this will include your full peer review and any attached files.

Reviewer #4: **Yes: **Virginia Zweigenthal

---

## [Editor Report · Acceptance letter]

31 May 2024

PONE-D-22-22471R2 

PLOS ONE

Dear Dr. D'Ambruoso, 

I'm pleased to inform you that your manuscript has been deemed suitable for publication in PLOS ONE. Congratulations! Your manuscript is now being handed over to our production team.

Kind regards, 

on behalf of

Professor Sogo France Matlala 

Academic Editor

PLOS ONE